**Impacts of land use and topography on soil organic carbon in a Mediterranean landscape (north-**
**western Tunisia)**
Donia Jendoubi[1, 2], Hanspeter Liniger[1] and Chinwe Ifejika Speranza[2]
[1] Centre for Development and Environment (CDE), University of Bern, Bern, 3012, Switzerland
[2] Institute of Geography, University of Bern, Bern, 3012, Switzerland
*Correspondence to*: Donia Jendoubi (Donia.jendoubi@cde.unibe.ch)
**Abstract:**
This study evaluates the impact of land use and topographic units (slope and aspect) on SOC within
the Wadi Beja watershed in north-western Tunisia. A soil spectral library was set to assess the variation
in the SOC of 1440 soil samples from four land use types (field crops, permanent crops, forest, and
grazing land), three slope categories (flat, moderate, and steep) and two aspects (north- and south-
facing). For field crops, only one factor – slope – significantly affected SOC, with SOC content in north-
facing areas appearing to be higher in flat areas (0.75%) than in hilly areas (0.51%). However, in south-
facing areas, SOC content was also higher in flat areas (0.74%) than in hilly areas (0.50%). For permanent
crops, which were interplanted with field crops, the slope significantly affected SOC content, which
improved to 0.97% in flat north-facing and 0.96% in flat south-facing areas, scoring higher than hilly
south- and north-facing areas (0.79%). In the grazing land use system, both of the investigated factors
– aspect and slope – significantly affected the SOC content, which was significantly higher in flat areas
(north-facing: 0.84%, south-facing: 0.77%) than in hilly areas (north-facing: 0.61%, south-facing: 0.56%).
For the forest, none of the factors had a significant effect on SOC content, which was higher in flat areas
(north-facing: 1.15%, south-facing: 1.14%) than in steep areas (1.09% in north-facing and 1.07% in south-
facing). This study highlights the ability of visible and near-infrared (VNIR) spectroscopy to quantify
C in diverse soils collected over a large diverse geographic area to indicate that calibrations are feasible,
and therefore, assessing the variation of SOC content under land use and topographic units (slope and
aspect) will result in better sustainable land management planning.
**Keywords:** soil organic carbon – land use – spectroscopy – topography – northwestern Tunisia
**1. Introduction:**
Land degradation is a major challenge for Mediterranean arid and semi-arid ecosystems (Hill et al.,
2008). In Tunisia, people are responsible for land degradation through deforestation, overgrazing,
removal of natural vegetation, and agricultural practices that erode soils (Sarraf et al., 2004). Long-term
anthropogenic pressure from agricultural use (Kosmas et al., 2015), in addition to abiotic factors such
as climatic variability and topographical variability (Scarascia-Mugnozza et al., 2000), create diverse
situations for which it is difficult to draw generally valid assumptions concerning soil organic carbon
(SOC) distribution and its determinant factors (Jobbagy and Jackson, 2000).
Soil degradation is a key component of land degradation in the Mediterranean region (Hartemink
2003), and soil quality deterioration contributes to the deterioration of other components of land
resources (e.g. water and vegetation) (Karamesouti et al., 2015).
Soil degradation processes include biological degradation (e.g. soil fertility and soil fauna decline),
physical degradation (e.g. compaction, soil erosion, and waterlogging), and chemical degradation (e.g.
acidification nutrient and depletion (Diodato and Ceccarelli, 2004; Post and Kwon, 2000), which are
caused by agricultural practices.
The soil quality concept has been proposed for application in studies on sustainable land management
(Doran, 2002). To measure soil quality, minimum data sets have been suggested that allow detailed
description by including soil chemical and physical indicators (Lal 1998). However, integrative
indicators are more appropriate for preliminary studies, as they efficiently provide insight into general
soil quality.
When using the term "soil quality", it must be linked to a specific function. In this study, soil quality is
seen in relation to soil conservation in agricultural systems, which aims to maintain the capacity of soil
to function as a vital living system for sustaining biological productivity, promoting environmental
quality, and maintaining plant and animal health (Doran and Zeiss, 2000).
Soil organic matter is one such integrative measure of soil quality, influencing soil stability, soil fertility,
and hydrological soil properties. OM plays a crucial role in soil erosion: when the erosion removes
surface soil, the OM and clay vanish, resulting in fertility decline, biological activity, and aggregation
(Wolfgramm et al., 2007). In soils with high calcareous silty amounts and in the absence of clay, OM is
particularly important with regard to the soil's physical properties (e.g. soil structure, porosity, and
bulk density), which again determine erodibility (Hill and Schütt, 2000).
Mediterranean soils are characterized by low amounts of OM, which results in a soil fertility decline
and structure loss (Van-Camp et al., 2004). Furthermore, SOC is variable across land use (Brahim et al.,
2010), and most agricultural soils are poor in OM, often comprising less than 1% (Achiba et al., 2009;
Parras-Alcántara et al., 2016; Muñoz-Rojas et al., 2012). In Mediterranean soils, loss of OM leads to root
penetration reduction, soil moisture, and soil permeability, which in turn reduces vegetation cover and
biological activity, and increases runoff and risk of erosion (Stanners and Bourdeau, 1995).
Tunisia has one the highest SOC depletion rates among Mediterranean countries (Brahim et al., 2010).
Its low soil fertility is considered a sign of its predominant inappropriate land management systems
(Hassine et al., 2008; Achiba et al., 2009). The soils from the study area are mostly derived from an
alteration of carbonate sedimentary parent material (marl, limestone, clay), cultivated under rainfed
conditions to produce cereal crops (wheat and barley). This form of cultivation decelerates the
mineralization of OM through a series of unsustainable practices including deep ploughing in spring
and summer, stubble ploughing in autumn to protect wheat against Fusarium, and various tillage
operations preceding sowing (Hassine et al., 2008). This relatively intensive soil cultivation,
accompanied by the practice of an annual application of phosphate and nitrogen fertilizers, is at the
root of the decrease in OM content following stimulation of microbial activity (Álvaro-Fuentes et al.,
73 2008).
Understanding the dynamics and SOC distribution as influenced by land use systems and topographic
features is critical for assessing land use management planning (Kosmas et al., 2000). SOC distribution
is influenced by topographic factors and climate variation, specifically temperature and water.
Currently, the northwest region of Tunisia is enduring extensive field crop monoculture and land
degradation owing to population increase, inappropriate land management, and rough topographic
formations. Much of the cropped land is unsuitable for agriculture and degrades quickly. The impacts
of agricultural practices and topography on nutrient cycling and ecological health, however, have not
been studied extensively in the Tunisian northwest region.
Due to this dispute, soil carbon levels measured through time can establish the long-term productivity
and possible sustainability of a land use system. In a nutrient-poor system, soil organic carbon (SOC)
can play an important role in the stability, quality, and fertility of the soil. Farmers and land use
planners are therefore interested in land use management that will enhance soil carbon levels.
In this study, we explore SOC distribution according to land use across topography (slopes and
aspects). The aim of this study is to quantify SOC content and evaluate the factors that affect SOC
variation, specifically the mechanisms affecting differences in SOC distribution patterns along different
land use systems and topographic units (slope and aspect) in a Mediterranean ecosystem dominated
by agricultural activities.
Information on soil quality is crucial for improving decision-making around efficient support of
sustainable land management. Thus, methods are needed to allow fast and inexpensive prediction of
important soil quality indicators such as SOC. The potential of diffuse reflectance spectroscopy in the
visible and near infrared (VNIR) range for fast prediction of soil properties in a non-destructive and
efficient way has been demonstrated by a number of studies (Amare et al., 2013; Shiferaw and
Hergarten, 2014; Shepherd and Walsh, 2002).
We are not aware of any study evaluating the impacts of topographic factors (slope and aspect) or
existing land use systems on SOC dynamics in Mediterranean agricultural soils, specifically in Tunisia,
based on an accurate and consistent database such as a soil spectral library.
Most soils are exposed to water erosion, which is provoked by poor cover cultivation practices and hilly
topography. The Wadi Beja watershed was selected because it comprises a variety of degraded areas
and areas where soil and water conservation practices (SWC) are applied. It is the most productive and
extended cereal area in Tunisia, and faces serious risks associated with monoculture production of field
crops under inappropriate land management practices. Some new practices, such as agroforestry, were
introduced into the region in the 1980s, along with permanent crops such as olive and almond trees.
The first research objective was to build a soil spectral library in order to apply it in the Wadi Beja
watershed, as there was no accurate or valid soil database for the studied region or even for the whole
country. The second objective was to examine the distribution of SOC under the different slopes,
aspects, and land use systems. The third objective was to investigate, specifically, three research
questions: (1) How does SOC vary under cereal monoculture and then after interplantation of
permanent crops? (2) How and why are ecosystems more sensitive to soil degradation (SOC loss) on
steep and south-facing slopes than on gentle and north-facing slopes? (3) How can land management
practices under different abiotic factors (e.g. topography) influence soil organic carbon (SOC) variation,
and what practices are recommended in this case study?

**2. Materials and Methods:**
**2.1. Study area**
The study area, the Wadi Beja watershed, lies at 36°37'60" N and 9°13'60" E in north-western Tunisia.
Upstream of Wadi Beja is the Amdoun region, and downstream the junction with Wadi Medjerdah in
the Mastutah region. Wadi Beja is a tributary of the Wadi Majerdah, the most important river in Tunisia
(figure 1).

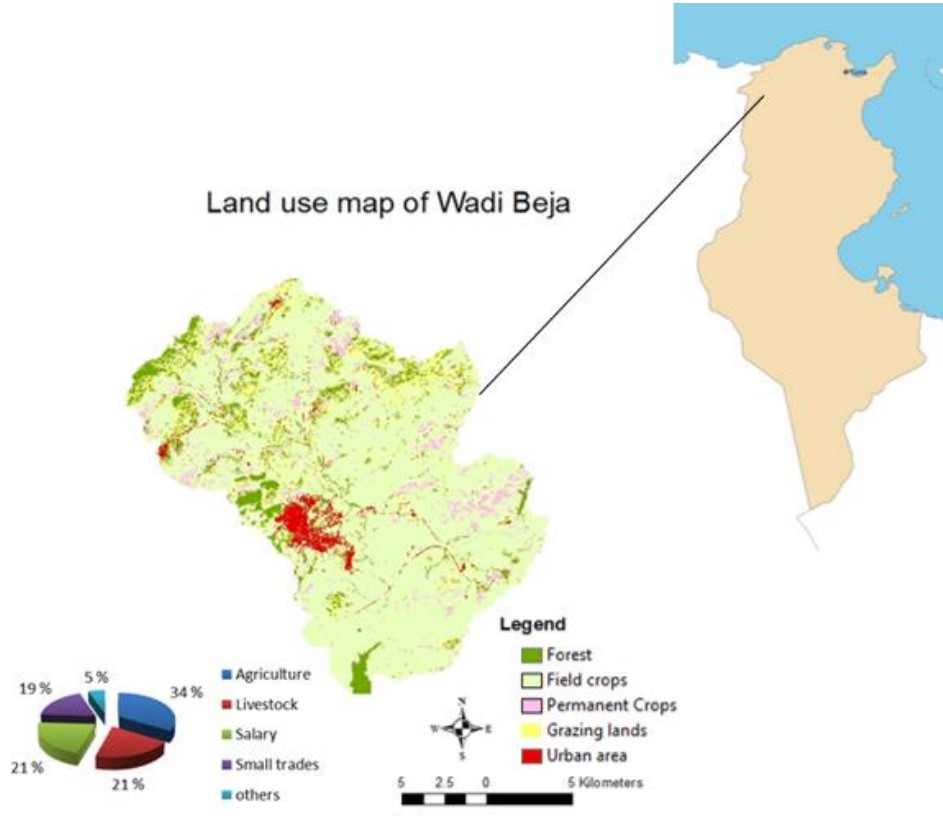


**Figure 1.** Characterization of household income, location and land uses of the study area, Wadi Beja watershed, northwestern Tunisia. Source: Jendoubi et al., 2019

The watershed (about 338 km²) covers diverse topographic environments, with an elevation ranging from 110 m a.s.l to nearly 750 m a.s.l; 64% of the surface has a high to steep slope and 36% has a moderate slope. Annual rainfall is irregular and varies from 200 mm to 800 mm. Early October to the end of April (late autumn to early spring) are considered the rainy seasons (AVFA, 2016). During the summer it is very dry and hot. The maximum temperatures are recorded at the end of July and range from 38°C to 44°C. Minimum temperatures are recorded at the end of December and fall between 6°C and 8°C (AVFA, 2016). In the Beja region, the population is mainly rural (56%), with 48.5% active in the agricultural sector. Agriculture remains the main source of household income (55%, including livestock) (figure 1). Nearly 78% of rural households live entirely off their farms (AVFA, 2016). There are three types of farming systems: extensive (83%), intensive (6%), and mixed (11%). Five different land use systems (LUS) have been defined: field crops (71%), grazing lands (10%), forest (9%), permanent crops (7%), and built-up areas (3%).

The current soil types in the study area are vertisols, which cover 46% of the total area, isohumic soils
(23%), brown calcareous soils (12%) and regosols (10%). Rendzinas soils, lithosoils, hydromorphic soils
and fersiallitic soils exist, covering small areas that add up to less than 9% according to the agricultural
map database of Tunisia.
Land management in the study area is similar in relation to land preparation, organic amendments,
crop rotation, and mulching (stubble, roots). Mineral fertilizers have been applied for several decades,
and cropland – the major land use – has been used for monoculture of cereal crops such as wheat and
barley.

**2.2. Methods**
**2.2. 1. Land use change history**
A land use system (LUS) is defined as the sequence of goods and services obtained from land, but can
involve particular management interventions undertaken by the land users as well. It is generally
determined by socio-economic market forces, as well as the biophysical constraints and potentials
imposed by the ecosystems in which they occur (Nachtergaele et al., 2010).
This study investigated four land use systems – field crops, permanent crops, forest plantation, and
grazing land – in order to assess their effects on the variation of SOC (table 1). Built-up areas and roads
were excluded. We used atmospherically corrected Landsat Surface Reflectance data images (4-5, 7 and
8) from 1985, 2002, and 2016 to derive the land use maps, in order to evaluate the changes over that
time period (Jendoubi et al., 2019).
The Landsat scenes were selected from among all those available in the green season (out of harvesting)
for the corresponding years; we considered only those with less than 20% of cloud cover overall and
without clouds on the study site area. Unsupervised classification was carried out for the images in
order to define the major land use systems. Following this, a validation based on ground truth data was
made in order to confirm the generated land use maps and assess their accuracies.

**Table 1.** The five major land use and management classes studied in the Wadi Beja watershed,
Tunisia

| Aggregated land use classes | 1985 | | 2002 | | 2016 | |
|---|---|---|---|---|---|---|
| | % | km² | % | km² | % | km² |
| Field crops | 82.1 | 272.7 | 76.4 | 254.0 | 71.0 | 236.2 |
| Grazing lands | 9.3 | 30.9 | 10.2 | 33.7 | 9.7 | 32.2 |
| Forests | 3.9 | 13.1 | 7.7 | 25.6 | 8.9 | 29.6 |
| Permanent crops | 3.4 | 11.2 | 4.2 | 14.1 | 7.3 | 24.4 |
| Built-up areas | 1.3 | 4.5 | 1.5 | 4.9 | 3.1 | 10.0 |
| Total | 100 | 332.4 | 100 | 332.4 | 100 | 332.4 |

**Source:** Jendoubi et al., 2019
Table 1 illustrates substantial land use and land cover change (LULC) in the Wadi Beja watershed after
1980. Field crops constituted the predominant land use type, accounting for approximately 82% in 1985
and 71% in 2016. Plantation forest also increased from 3.9% in 1985 to 9% of the watershed in 2016. In
1980, to remedy the degrading effects of monoculture of annual cropping, deforestation, and
overgrazing on the pastures and the forests, a programme developed by ODESYPANO (Office
Development Sylvo-Pastoral Nord Ouest) and financed by the World Bank implemented some
conservation activities including development of permanent vegetative cover using olive trees and
sylvo-pastoral management. An agroforestry (agro-sylvo-pastoral) system was introduced in 1982 as
an alternative programme for development and conservation in the region. This system included
converting annual cropping into a combination of annual crops interplanted with olive trees (in this
study classified as "permanent crops"). This area increased from 3.4% in 1985, when it was introduced
for the first time in the region, to 7.3% in 2016. The local farmers took this alternative as they believed
that their soils had become poor and no longer gainful for annual crop production. Grazing land
remained almost unchanged in terms of area, as it is spread over badlands, barren lands, and riverbanks
with a high concentration of eroded and poor soils.

**2.2.2. Soil sampling**

We selected four land use systems (LUS) (excluding built-up areas), three slope classes, and two aspect
classes to study their interrelations and their effects on SOC. The LUS were forests, field crops,
permanent crops, and grazing land (table 1). Aspect and slope units were derived from Lidar DTM,
aligned, and resampled to 30m. Slope categorization was based on the FAO soil description guidelines
(Barham et al., 1997). The slope categories were grouped into three: flat, moderate and steep. Aspect
was categorized into two classes: north and south. Details about slope and aspect categories are
presented in table 2.

**Table 2.** Slope and aspect

| Slope (in %) | Aspect (azimuth degrees) |
|---|---|
| 0 to 8 (Flat) | 0 to 90, 270 to 360 (North) |
| 8 to 16 (Moderate) | 90 to 270 (South) |
| > 16 (Steep) | |

From all slope, aspect classes, and different land use systems (LUS), soil samples were collected
randomly from the topsoil (0-20 cm). In a factorial randomized design considering the four land use
types, the three slopes, and two aspects, a total of 24 different sampling units (n=4×3×2) were
considered. In total, 1440 soil samples were collected from all the sampling units in the topsoil layer (0-
20 cm) using a soil auger (10 cm diameter) with an average of 60 samples per sampling unit.

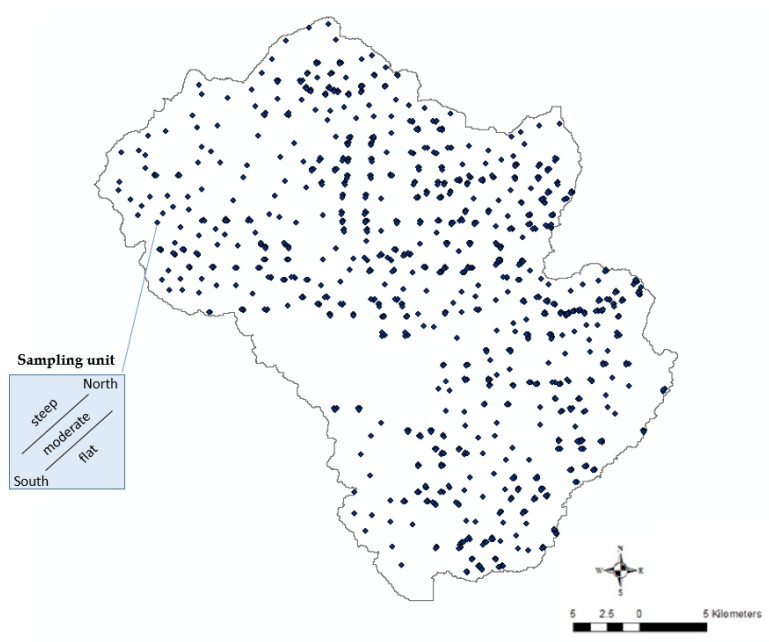

**Figure 2.** Location of the soil samples and the sampling design.
The sampling design shown in figure 2 summarizes the strategy of the sampling, where each soil
sample can be taken in a randomized way from any specific sampling unit. Sampling units are listed
as follows: field crops (flat north and flat south), field crops (moderate north and moderate south), field
crops (steep north and steep south), permanent crops (flat north and flat south), permanent crops
(moderate north and moderate south), permanent crops (steep north and steep south), forests (flat north
and flat south), forests (moderate north and moderate south), forests (steep north and steep south),
grazing lands (flat north and flat south), grazing lands (moderate north and moderate south) and
grazing lands (steep north and steep south).
**2.2.3. Soil analysis and spectral library**
The soil spectral library was set according to protocols cited by Shepherd and Walsh (2002), and
includes the following steps: (1) representative sampling of soil variability in the study area; (2)
establishing the soil reflectance spectral dataset using VNIR spectrometry; (3) selecting a reference
dataset to be analysed using traditional soil chemical methods required as reference values (450
samples, or 30% of the total, were selected according to their spectral variability); (4) determination of
SOC by means of soil chemical analysis (CNS elemental analysis); (5) calibrating soil property data to
soil reflectance spectra by applying multivariate calibration models; and finally (6) prediction of new
samples using the spectral library.

The soil spectral library for prediction of SOC was adjusted using a mug-light for illumination as described by Mutuo et al. (2006). Soil spectral reflectance was measured under standard conditions in the laboratory. Air-dried ground soil samples of 2 mm thickness were filled into borosilicate Duran glass Petri dishes with optimal optical characteristics. The Petri dishes were placed on a mug-light equipped with a Tungsten Quartz Halogen light source (Analytical Spectral Devices, Boulder, CO). Spectral reflectance readings were collected through the bottom of the Petri dishes using a FieldSpec PRO FR spectro-radiometer (Analytical Spectral Devices, Boulder, CO). Every sample was measured twice, with the sample rotated by 90 degrees for the second measurement. The two measurements were averaged, which minimized light scatter effects from uneven particle size distribution on the Petri dish floor. The instrument works with three spectro-radiometers to cover the wavelengths from 350 to 2500 nm at an interval of 1 nm. The fore-optic view was set to 8 degrees. For dark current readings, 25 scans were averaged, while for white reference and soil spectral readings 10 scans were averaged by the spectro-radiometer. Before each sample reading, white reference readings were taken from a spectralon (Labsphere) that was placed on a trimmed Petri dish bottom.

Pre-processing of soil reflectance data to decrease the noise present in the data and thus increase robustness of reflectance spectral data is common in VNIR spectrometry, and is especially important in the case of measuring setups that are difficult to control (e.g. due to power fluctuations or different operators during different measuring sessions). The main pre-processing steps conducted were as follows: Spectra were compressed by selection of every 10th nm. Spectral bands in the lowest (350-430 nm) and highest (2440-2500 nm) measurement ranges were omitted due to a low signal to noise ratio (lower than 90). The final number of wavelengths used as model input was 205. Information for these 205 wavelengths was further processed: the instrument covers the full wavelength range with three spectro-radiometers. Steps in the spectral reflectance curves were observed at the spectrometer changeovers. Most likely, this effect resulted from the Petri dishes used as sample holders and their specific index of refraction.

When choosing the validation set, care was taken to assure that validation samples were representative for the whole study area. Thus, samples were systematically chosen by selecting from every land use system and under the different (slope and aspect) sampling units. These samples, which constituted 30% of the total samples, were selected for chemical analysis, which was used to validate SOC model prediction.

Regarding the chemical method, the elemental CNS analyser (vario micro cube) was used for SOC estimates. For SOC measurement, 1 g soil is pre-treated with 10 drips of H3PO4 in order to remove carbonate. The sample is combusted at 1150°C with constant helium flow, carrying pure oxygen to ensure complete oxidation of organic materials. The $CO_2$ gas is produced and detected by a thermal

conductivity detector. Total soil carbon is measured, using the same procedure without pre-treatment
with H3PO4. Soil inorganic carbon is calculated as the difference between total soil carbon and SOC.
A calibration and validation with Partial Least Square Regression were used based on cross-validation
("leave one out") in order to ensure simultaneous reduction/correlation of both the spectral
information and the concentration data obtained from the chemical analysis.
After prediction of the remaining SOC sample values, a set of statistical parameters was applied in
order to assess the accuracy of results such as: the coefficient of determination ($R^2$), which measures
how well a regression line estimates real data points; the Residual Prediction Deviation (RPD), which
evaluates the quality of a validation; and the Root Mean Square Error of the prediction (RMSEP), which
assesses the accuracy of the model. These parameters evaluate the performance quality of the soil
spectroscopy model (Rossel et al., 2006).
A script for running various steps was made in RStudio in order to generate the soil spectral library.
**2.2.4. Statistical analysis**
Regarding the soil spectral library analysis, the partial least squares regression (PLS regression) was
used in RStudio to validate the spectral prediction model while assessing the coefficient of
determination (R2), residual prediction deviation (RPD), and root mean square error of the prediction
(RMSEP).
After generating the soil spectral library, a test of normality based on Sullivan and Verhoosel (2013)
was carried out in order to check the normality distribution of the data.
The Statistical Package for the Social Sciences (SPSS 20.0) software was used in order to compare the
averages obtained under the different factors. Variance analyses and multiple comparisons (MANOVA
test) were carried out to determine the effect of the different factors (land use, slope, and aspect) on the
variation of the SOC. Results were significant when $p < 0.05$. The interaction effect between the factors
was tested using the technique of split file. The results were grouped according to the land use factor
and the effect of the slope and aspect were tested in each land use value.
Results were presented in histograms using Excel XLSTAT. We then assessed the variation of SOC
under the different selected factors.
**3. Results**
**3.1. Soil spectral library as an integrative indicator of soil quality**
SOC content was plotted against SOC content predictions as displayed in figure 3.

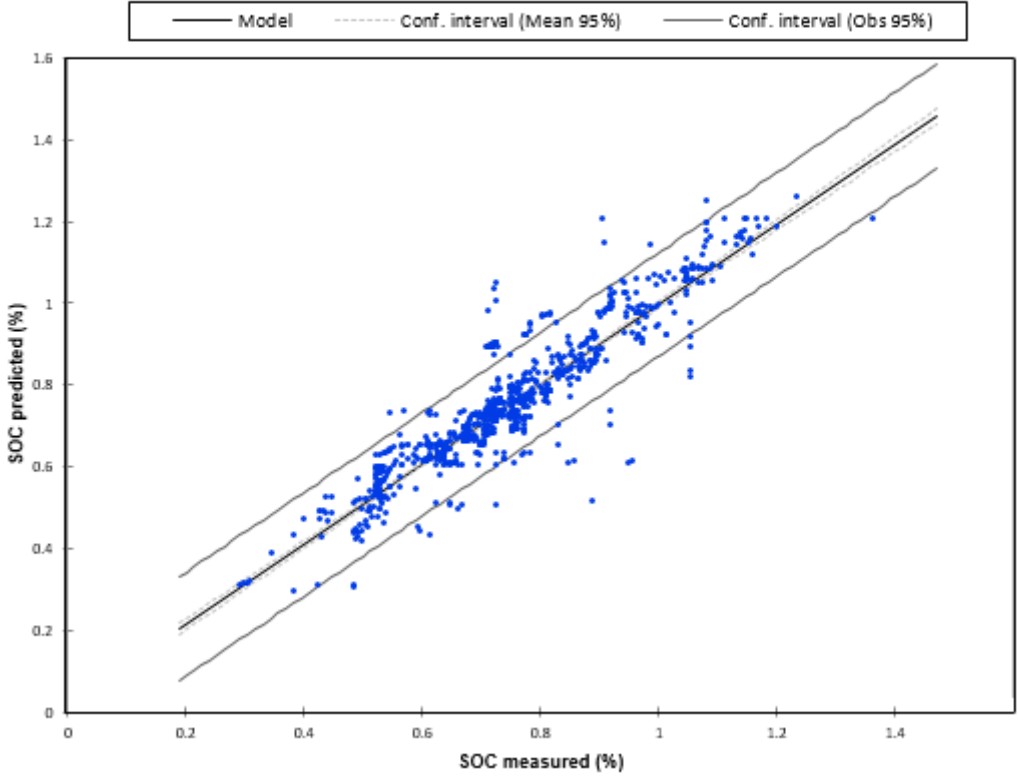

**Figure 3:** SOC values from chemical analysis plotted against SOC prediction.

The obtained spectral prediction model has R2= 0.85, RPD= 2.11, and RMSEP= 0.35%, which was rated excellent for prediction because RPD>2 (Viscarra Rossel et al. 2006). This means that the model is able to determine accurately the SOC content of 85% of the samples. The RPD (2.11>2) also shows that the model developed is of good quality and can be used to predict the remaining spectra and for further development of the spectral library.

Regarding the normality of the data, the test shows a high correlation of 0.95 between the overall data and their corresponding z-scores. Therefore, this means that the data are approximately normally distributed.

**3.2. Significance effects of all the variables**

A multivariate MANOVA analysis revealed which variables had statistically significant differences in SOC related to land use systems, slopes, and aspects. Table 3 reveals the results of the significance analysis for each of the three variables. The highest significance was reported for land use, followed by slope and aspect.

**Table 3.** MANOVA results showing the significance of the impact of land use, slope, and aspect for soil organic carbon (SOC) (n= 1440)

|  | F | Sig. |
|---|---|---|
| **LUS** | 395.263 | **0.000** |
| **slope** | 76.505 | **0.000** |
| **aspect** | 11.093 | **0.001** |

Sig. < 0.05 (statistically significant difference), in bold.
Sig. > 0.05 (no statistically significant difference)

The analysis of the significance of the different variables for each land use type is presented in table 4.

**Table 4.** MANOVA results regarding significance of all the variables under different LUS.

| LUS | Variables | F | Sig. |
|---|---|---|---|
| **Forests** | slope | 1.806 | 0.176 |
|  | aspect | 2.931 | 0.094 |
| **Field crops** | slope | 51.429 | **0.000** |
|  | aspect | 1.028 | 0.312 |
| **Permanent crops** | slope | 36.474 | **0.000** |
|  | aspect | 0.068 | 0.795 |
| **Grazing lands** | slope | 8.242 | **0.001** |
|  | aspect | 5.971 | **0.017** |

Sig. < 0.05 (statistically significant difference), in bold.
Sig. > 0.05 (no statistically significant difference)
For forest land use, no variables were significant, indicating that the variation of the SOC with high amounts in those components was not related to slope or aspect. For field crops and permanent crops, only slope had a significant effect on SOC. For grazing lands, both variables (slope and aspect) revealed significant effects on SOC content.

**3.3. SOC according to land use systems**
SOC content for different land use systems is shown in figure 4. The forest LUS had the highest SOC
content, with 1.09%. Permanent crops had the second highest values with 0.87% of SOC. The lowest
SOC content was found for field crops (0.70%) and grazing soils (0.74%).

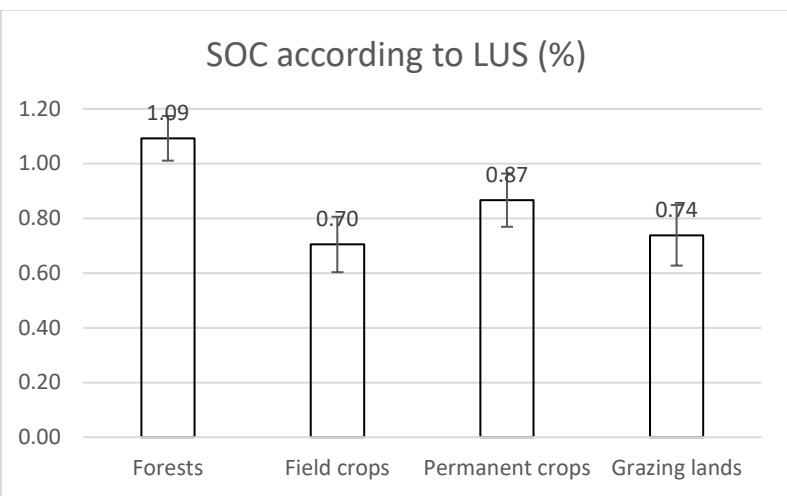


321         **Figure 4.** SOC rates according to land use systems in the Wadi Beja watershed, Tunisia.

According to the MANOVA results, land use systems significantly affect SOC content. In the study
area, the lowest SOC content was found in field cropping soils (0.70%), and the highest SOC content in
the forests (1.09%).
**3.4. Impact of slope and aspect on SOC**

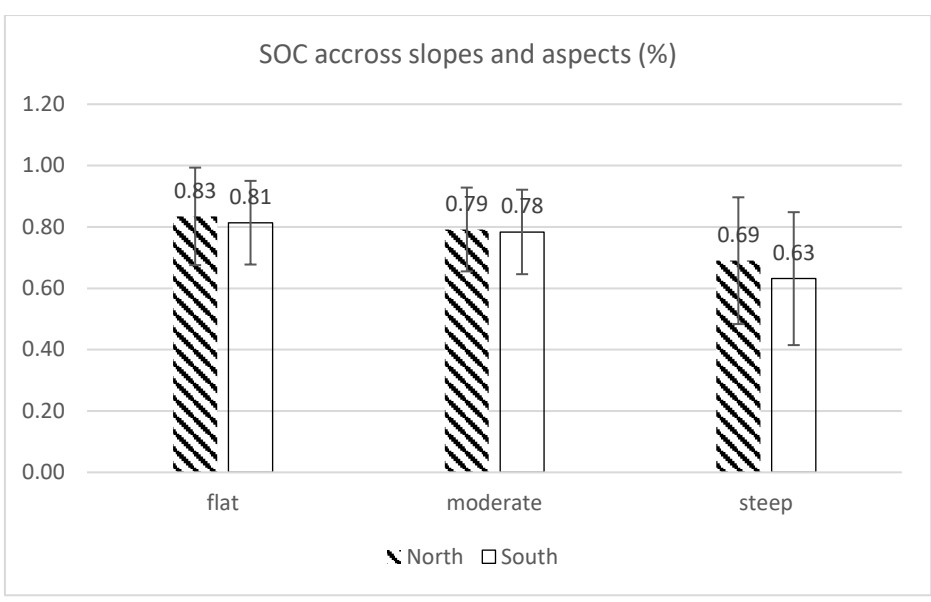


**Figure 5.** SOC rates according to slope and aspect in the Wadi Beja watershed, Tunisia.
Figure 5 shows the highest SOC content (0.81%-0.83%) on flat slopes and slightly reduced SOC on
moderate slopes (0.98%-0.79%). Both flat and moderate slopes show no significant difference between
northern and southern slopes (difference <0.02%). The lowest SOC was on steep southern slopes with
0.63%, followed by steep northern slopes with 0.69% SOC.
**3.5. Impact of land use, slope, and aspect on SOC**

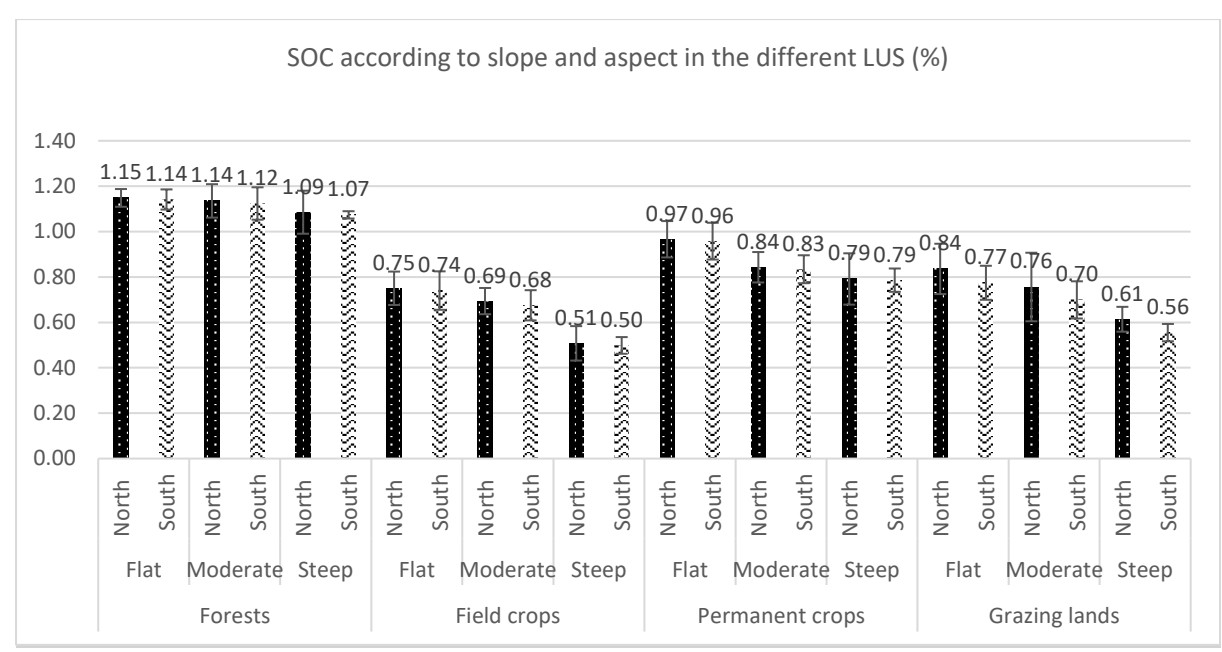

**Figure 6.** SOC rates according to slope and aspect for the different land use systems.

When evaluating the impact of slopes on SOC variations under the different LUS, the results presented in figure 6 reveal that in forest plantations, the highest SOC amounts were observed in flat (1.15%) and north-facing areas. 1.14% SOC was found on moderate slopes in north-facing areas, and 1.09% on steep north-facing areas. As previously shown, statistically, the slope has no significant effect on SOC variation under the forest LUS.

For field crops, the highest SOC content was found in flat north-facing areas (0.75%), followed by 0.69% on moderate slopes in north-facing areas and then very low figures of 0.51% on steep slopes in north-facing areas and 0.50% in south-facing areas. Figure 6 clearly shows a marked decline in SOC with increasing slopes under field crops. For permanent crops, the decline with increasing slopes is less than that of the field crops, and the SOC for all slopes are increased. The highest SOC content was found in flat north-facing areas (0.97%), followed by 0.84% in moderate areas and 0.79% in steep areas. Finally, on grazing lands, the different slopes show marked differences, with 0.84% of SOC in flat north-facing areas, 0.77% in flat south-facing areas, 0.76% in moderate north-facing areas, 0.7% in moderate south-facing areas, 0.61% in steep north-facing areas, and 0.56% in steep south-facing areas.

The MANOVA test shows that aspect has no significant effect on SOC variation for forests, field crops, or permanent crops. Only for grazing land does aspect have a significant effect on SOC variation, with north-facing soils having a greater SOC level than south-facing areas. See figure 6 and table 4.

## 4. Discussion

Regarding the soil spectral library, comparing the results from the study carried out by Hassine et al. (2008), which concluded that SOC content does not exceed 2% in north-western Tunisia, our prediction model falls within this amount with a maximum organic carbon percentage of 1.2%. This state of low OM in soils used for agriculture, compared to forests with little indication of soil degradation, has been confirmed by various authors (Arrouays et al., 1994; Cerri, 1988; Robert, 2002). This low content has negative impacts on the soil structure, which is built mainly by means of mineral colloids and whose stability is affected, leading to numerous deficiencies in production and susceptibility to degradation factors. Cereal soils may have acquired a balance between SOC inputs and losses, but at a very low equilibrium level compared to forests; the latter have less decline of SOC and have been protected against erosion, which is the main type of land degradation in the study area (Hassine et al., 2008).

Previous studies show that SOC can play a significant role in monitoring soil quality related to land use and reduction of soil degradation (Shukla et al., 2006; Hassine et al., 2008). We thus focused on the SOC content, which we calibrated from soil reflectance spectra. Chemical analysis of SOC made it possible to calibrate and validate a model using soil spectra to predict a wider range of soil samples. The soil spectral library of 1440 soil samples was used to investigate SOC under the various combinations of land use, slopes, and aspects. The library made it feasible to gain some interpretations and assess SOC variation in the study area, and therefore to generate some recommendations for land use planners.

The results on the impacts of land use on SOC indicate that field crops have the lowest SOC content. This could be the result of land degradation due to inappropriate agricultural management such as intensive tillage, the removal of crop residues, reduced vegetation cover, deteriorated soil aggregation and erosion, and a continuous monoculture system. This finding is coherent with the results of several researchers (Lemenih and Itanna, 2004; Lal, 2005; Muñoz-Rojas et al., 2015; Hamza and Anderson, 2005) who have revealed a significant decline in OM content in cropland compared to natural forests. Herrick and Wander (1997) found that in annual cropping systems, the distribution of SOC is highly influenced by land management practices such as reduced tillage, rotation, fertilization, and shifting cultivation. Consistent with the study by Hassine et al. (2008) in northwestern Tunisia, the reduced OM decomposition rates are a result of intensive agricultural practices; monoculture, tillage on steep slopes, and tillage in wet seasons, in addition to other topographic factors, may lead to a decrease in SOC.

Changing annual field crops by interplanting them with permanent tree crops has increased the SOC of soils under previous annual field crops almost halfway to the level of forests (0.87%). Intercropping previously mono-cropped fields with tree crops (olive, almond, and pomegranate trees) between 1982 and 1985 significantly improved the SOC within 30-35 years. Creating agroforestry systems in this way

is considered to have been a good land management intervention in northwestern Tunisia, as it reduced
both the area covered only by very old cereal monocultures and the soil degradation. However, some
farmers made no changes to their land management, as they did not perceive the advantages of the
agroforestry system (Jendoubi and Khemiri, 2018). Yet agroforestry systems are globally recognized as
having a high potential to sequester C, since they are more accomplished at capturing and utilizing
resources than grassland systems or single-species cropping (Nair et al., 2011).
Grazing lands, even though they are not tilled, have a low level of SOC (0.74%), only slightly higher
than annual crops (Figure 3). Continued overgrazing and reduction of vegetation cover seem to
degrade the soils and their SOC. A low SOC content can continue due to a lack of appropriate grassland
management. Open pasture without canopies and weak grass-vegetation cover increase the
vulnerability of this land use system to soil degradation and SOC decline. Various studies have shown
that the way grazing land is managed affects SOC (Wu et al., 2003; Soussana et al., 2004): overused
grazing lands with less vegetation cover are more affected by soil erosion and soil exposure to wind
and rain, leading to greater SOC loss. Notably, grassland management strongly affects SOC stocks,
which decrease as grazing intensities increase (Neff et al., 2005).
The highest SOC amounts were found in the forests. The explanation for this is that forest has a dense
cover that protects soil from being exposed to any other factors such as erosion, and the SOC cannot be
affected. It can be assumed that the soil under forest has no degradation caused by soil erosion from
water, as observed in some surrounding fields. This finding has been confirmed by many authors who
have shown that in Mediterranean areas, many forest soils are rich in OM; as a consequence, these soils
supply a large quantity of carbon, which means that they are distinguished by high SOC (Lal, 2005;
FAO, 2010), which is highly related to the lower disturbance in the forests.
The Mediterranean region is generally characterized by poor soils with low OM content (around 1%)
due to their nature and to being overused by agriculture, which means that they have low C inputs
from plant residues and low canopied density, and are subjected to inappropriate management
practices (Verheye and De la Rosa, 2005; Cerdà et al., 2015).
Land management is shown to be a key indicator affecting SOC distribution, influencing topsoil in
particular (Ferreira et al., 2012). In Mediterranean areas in particular, land management is a significant
factor given the limitations to SOC accumulation. Moreover, high SOC reflects undisturbed soil and
high soil quality, as is the case in forest land use (Corral-Fernández et al., 2013).
The interpretations emphasize that the impacts of land use on SOC variation is highly related to land
management practices. The findings highlight the contribution of overuse and monoculture to SOC
decline under the field crops land use system. In lands where field crops once were, if they have been
interplanted with permanent crops, the SOC content has improved. Overgrazing and bad management
of grazing lands has led to SOC decreases. Finally, forest land use has the highest SOC content, as it is
protected by forest regulation and less disturbed.

With regards the impact of slope on SOC variation, our results show that the higher the slope, the lower the SOC content. Irvin (1996) specified that generally, with increasing slope, OM lixiviation is reduced, mineral is weathered, clay is translocated, and horizons are differentiated.

Moreover, topographic position has a significant impact on soil temperature, soil erosion, runoff, drainage, and soil depth – and hence soil formation. Different soil properties encountered among topographic units (slope and aspect) will affect the litter production and decomposition, which will undoubtedly have effects on SOC content. The accumulation of SOC variation on hillslopes is explained by the decomposition rates of OM and litter input differences (Yimer et al., 2006).

When assessing the results of the impact of aspects on SOC variation, south-facing terrain has lower SOC content than north-facing terrain, which is explained by its exposure to the highest solar radiation and, in particular, the highest temperature during the vegetation period and the long hot summers. This implies high evaporation and a high burn down of OM due to high temperature, less moisture in the soils, and consequently a slow-down of the decomposition of OM.

In addition, according to our findings, the impact of both slope and aspect on SOC content was very distinct, as indicated statistically by a significant effect on SOC content in the MANOVA. The issue is that steep and south-facing slopes are more sensitive to degradation than other areas, which is explained by the fact that steepness increases runoff and soil erosion, and southern exposure increases evapotranspiration and temperatures, thus decreasing the availability of nutrients, water, and SOC to plants. Apart from differences in land use management, SOC variation is mainly affected by environmental factors in soil along with topographic units (slope and aspect).

The literature links temperature and moisture to OM decomposition in soils (García Ruiz et al., 2012; Griffiths et al., 2009). As shown by Garcia-Pausas (2007), in the Mediterranean area, shaded areas such as northern-facing or colder southern areas sustain regularly high moisture content for longer and consequently become more fertile and productive, in contrast to the southern-facing areas that are exposed to high radiation and thus occasional water deficits.

With regard to steepness and aspect, the higher the slope, the more exposed to the south, and the more affected by erosion and different climatic conditions, the lower the SOC content (Yimer et al., 2007; Yimer et al., 2006). Different topographic positions are considered to have different microclimatic and vegetation community types and thus significant variations in SOC. Topography (slope and aspect) hence plays a crucial role in relation to temperature and moisture regimes. The temperature is highly influenced by solar radiation, which has a role in soil chemical and biological processes and vegetation distribution (Bale et al., 1998). Hence, the temperature of the soil plays a key role in monitoring the biomass decomposition rate, and thus affects the SOC distribution, either delaying or accelerating its decomposition (Scowcroft et al., 2008).

From the results assessing the impact of slope combined with land use, we can see that the highest SOC content was observed in the flat area under all land use systems, and it tended to decrease in steep positions. In general, under all land use systems, we can observe the same tendency of SOC variation, ranging from highest SOC content in flatter positions to lowest in steep positions.

This can be explained by the fact that soils on flat slopes tend to be thicker as a result of deposition.
Erosion causes stripping of the soil in hillslope areas. As shown by Yoo et al. (2006), the prevalent
portion of SOC is deposited in depositional areas, with hillslopes being more susceptible to sporadic
mass wasting events, continuous soil erosion and production, and consequently less SOC storage. In
addition, the highest erodibility is related to hilly areas where soils have a tendency to be shallow,
coarse in texture, and low in OM, while lower erodibility is observed in flat areas with organic-rich,
deep, and leached soils (Lawrence, 1992).
From the clear difference in the variation in SOC under forest and field crop land use systems, we
interpret that it is the land use factor that dominates SOC distribution rather than the slope factor.
In general, steep slopes have a lower SOC content than flat land, as they are more vulnerable to erosion,
especially when associated with inappropriate management and overuse (Reza et al., 2016; Bouraima
et al., 2016). Cropland in sloping areas is highly vulnerable to water erosion, which leads to extensive
soil disturbance, while land use patterns affect vegetation cover, soil physical properties such as SOC,
and surface litter. Therefore, this provokes the runoff and soil erosion processes that accompany
nutrition loss (Dagnew et al., 2017; Montenegro et al., 2013). Therefore, the extent of nutrition loss
differs according to land use systems, as is the case with cereal monoculture in the study site.
Therefore, under different land use systems, the difference in SOC content is related to the effect of
variation in land use system intensity along the toposequences. As shown by our results, higher SOC
content was recorded in the forest where there is less disturbance and use, and statistically slope has
no significant effect on SOC variation. In the field cropping area, the fact that soils are overused and
subject to continuous intensive cultivation without appropriate soil management practices has
contributed to the degradation of important soil quality indicators such as SOC. Hence, in order to
improve and maintain soil quality parameters for sustainable productivity, it is crucial to reduce
intensive cultivation and integrate the use of inorganic and organic fertilizers.
In agricultural areas, continuous intensive cultivation without appropriate soil management practices
has contributed to loss of SOC. Kravchenko et al. (2002) and Jiang and Thelen (2004) found that within
variability in topography, slope was considered to be a major crop yield limiting factor.
Correspondingly, after plantation of permanent crops in combination with field crops, SOC content
was enhanced, in keeping with the tendency for the highest SOC content to be in flat areas and the
lowest in steep areas. Herrick and Wander (1997) showed that after introducing permanent crops, slope
significantly affected SOC content.
According to the obtained results on the impact of all factors on SOC variation, with less SOC revealed
in south-facing areas, our finding lends strong support to the interaction effects of slope and aspect on
OM decomposition (Griffiths et al., 2009), as the difference between north- and south-facing areas is
solar radiation, wind, and rainfall.
According to McCune and Keon (2002), the reason for these results is that slope and aspect play a
significant role in solar radiation redistribution, hence the solar radiation heterogeneity on hillslopes
leading to differences in soil moisture and temperature. Huang et al. (2015) stated that the SOC
concentration in shaded aspect areas was significantly higher than in sunny aspect areas. Therefore, as
discussed previously, increases in SOC and OM accumulation are generated by means of increased
moisture and reduced temperature. Decreased soil temperature usually results in decreased OM
decomposition rates and litter decay rates (Blankinship et al., 2011).
For grazing lands, all the variables (slope and aspect) revealed significant effects on SOC content as also
shown in the study of Bird et al. (2001). SOC content is generally low, though it is higher in flat areas.
This is explained by overgrazing and pressure in the different topographic positions, as they are all
easily accessible to livestock. Even on steep slopes there is pressure and overgrazing, in addition to the
exposure of these areas to erosion by wind and rain. This may reveal the vulnerability of this land use
system to erosion and deterioration of soil quality.
Why are grazing land use systems the most sensitive to all the tested variables (slope and aspect)? This
can be explained by the fact that in the case study, grazing land was generally open grassland and it is
evident that soils are more sensitive in open grassland than under tree canopies, as SOC stocks under
tree canopies are in general higher than in open grassland (e.g. Seddaiu et al., 2013). Moreno et al. (2007)
stated: "The amount of SOC in the topsoil beneath the tree canopies projection was around twice as
high as beyond the tree canopy". This can also be related to overgrazing, as shown in a literature review
of the effects of overgrazing in the Mediterranean basin (Sanjari et al., 2008; Costa et al., 2012).
Furthermore, the semi-arid climate and inclined topography prevailing in the Mediterranean grazing
lands render ecosystems vulnerable to SOC losses. As shown by Ryan et al. (2008), the higher the level
of grazing, or the greater the residue removal, the greater the decline in mean OM level. The reason
behind the decrease in carbon and nutrient cycling is mainly that OM in grassland is accumulated in
roots, which leads to its loss with every removal of aboveground biomass.
The most likely clarification for the results obtained on decreased SOC content in steep south-facing
areas under the field crops land use system is that soils are affected by soil degradation initiated by
inappropriate land management and consequently have weak vegetation cover. This condition makes
these soils more sensitive to the south-facing exposition characterized by higher solar radiation and
evaporation, and thus decreases soil moisture, biological activity, and SOC loss. Wakene and Heluf
(2004) have also indicated that intensive cultivation aggravates OM oxidation and hence reduces OC
content.
Therefore, some options for sustainable land management practices can be recommended, such as
establishment of enclosures (Mekuria and Aynekulu, 2013), which could be efficient in recovering the
degraded grazing land areas of the watershed. In addition to protecting trees against damage caused
by uncontrolled grazing animals by installing fences and trunk protection, mixing of animal species
(mostly sheep and goats, but also cows and horses and setting additional fodder provision could be a
feature of the summer season.
In order to maintain improved soil quality and sustainable productivity in cropping lands, there is a
need to reduce intensive cultivation, agroforestry, and practice of fallow, integrate use of inorganic and
organic fertilizers, and pay more attention to the most vulnerable areas (steep and south-facing areas).
There are strong indications that agroforestry has been successful in retaining and even improving SOC
and soil fertility: the results show that introducing an agroforestry system – e.g. combining an olive
plantation with annual field crops – has increased SOC content in the most vulnerable areas. Thus, such
types of sustainable land use should be the focus of land managers and land use planners.

## 5. Conclusions

Land management can profoundly affect soil C stocks and appropriate management can be used to
sequester soil C. As with all human activities, the social dimension, especially land management, needs
to be considered when implementing soil C sequestration practices. Since there will be increasing
competition for limited land resources in the coming century, soil C sequestration cannot be viewed in
isolation from other environmental and social needs. In order to increase soil C sequestration, as part
of wider plans to enhance sustainable land management, more attention should be paid to the
importance of land use and the different topographic factors. In areas with exceedingly erodible soils,
such as those on steep slopes and south-facing zones as shown in this study, application of soil and
water conservation measures is crucial to sustain agricultural fields and prevent or reduce soil
degradation. Greater efforts are required on steep slopes and south-facing land to reduce SOC decline
than in flat areas and north-facing land. However, further study of the areas is recommended, especially
in combination with other topographic factors such as altitude and curvature.
By far the best option, however, is to identify land management practices that increase C stocks whilst
at the same time enhancing other aspects of the environment, e.g. improved soil fertility, decreased
erosion, greater profitability, or improved yield of agricultural and forestry products. There are a
number of management practices available that could be implemented to protect and enhance existing
C sinks now and in the future.
Since such practices are consistent with, and may even be encouraged by, many current international
agreements and conventions, their rapid adoption should be as widely encouraged as possible.
Finally, this paper contributes towards filling a gap in investigation on the impacts of various land uses
on SOC in Tunisia. The results presented in this paper are valid for calibration of further soil spectral
libraries in northwest Tunisia; this was the first soil spectral library collated in Tunisia and the
methodology can be replicated and applied to other areas. Further studies on SOC variation depending
on land use and topographic factors are needed to inform sustainable land management in Tunisia.

## 6. Acknowledgments

This research was funded by the Islamic Development Bank, Grant/Award Number: 78/TUN/P34 and
the Swiss Federal Commission for Scholarships for Foreign Students, Grant/Award Number:
2014.0968/Tunesien/OP. The authors thank Tina Hirschbuehl, Dee Cooke, and William Critchley for
English proofreading.

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
