# Peer review of "Impacts of land use and topography on soil organic carbon in a Mediterranean landscape (north- western Tunisia)"

_SOIL, 2019_

## Referee Comment (RC1) · Anonymous Referee #1 · 3 May 2019

GENERAL COMMENTS The article discusses a very important topic and is based on extensive dataset. However, because of poor English and lack of logic in presentation of research, the reading is quite difficult. Moreover, there is a lack of adequate explanation of the research context (state-of-the-art revision and indication of the scientific gap to be approached and methodology. We would recommend the author to look at the recent review of the topic by Thangavel Ramesh et al. (Advances in Agronomy, 2019). What is the new knowledge this work contributes to that already published? The article considers "land forms" as synonym to slope and aspect, which is not true. There is no clear explanation of methodology; confusion in Results and Discussion sections, and, as a consequence, Conclusions are not supported by Results and Discussion. Incor-

rect citing format through the article (you cannot begin the sentence with parenthesis). Example: Line 210: "(Kravchenko et al., 2002 and Jiang and Thelen, 2004) found . . ." Line 212: "(Herrick and Wander, 1997) showed . . ."

Specific comments: English style is very poor. The article is full of English mistakes of all types. Some examples: Abstract (lines 13-14): "For permanent crops, which was interplanted . . .." – the verb was is incorrect because crops is a plural noun Line 35: ". . . soil quality was seen in relation to soil conservation in agricultural systems, which aims at sustaining the productive capacity of soils, . . ." – impossible Line 167: "The performance evaluation of the prediction model was created on the following statistical criteria: . . ." - impossible Line 211: "slope plays a great role on crop yield" – mistake in the use of preposition Line 217: "Forests showed significantly the highest SOC" – the weird word order

Methodology: No table with descriptive statistics of data. No details on data preprocessing, spectral measurements instrumentation and protocol. No explanation of classification procedure used for land use detection from Landsat images. The method applied for assessment of slope and aspect impact is not explained. It seems that only one-way ANOVA was used to assess the impact of slope and aspect on SOC. Application of multivariate statistical modeling would be more appropriate. One of the key experts in VIS-NIR spectroscopy is incorrectly cited two times (different errors): Line 171: "Vicsarra et al., 2006" Line 188: "Rossel et al. 2006)"

Figures:

Figure 1, the map should be improved, the gray backgound (topographic sheet) should be removed. Figure 5 and 6 contain redundant information.

---

## Referee Comment (RC2) · Anonymous Referee #2 · 7 May 2019

Soil-2019-15 is an interesting study which presents the combined effect of land use and topography on soil organic carbon (SOC) by comparing SOC data under four land uses and three slope categories and two aspects in north-western Tunisia. This paper contributes to build a soil spectral database of the studied watershed as a tool to study SOC variability. Although the consistent soil sampling collection and laboratory work, there is a lack of detailed information about materials and methods (soil type, soil spectral collection, model spectrometer, partial least-square regression) and the statistical analysis can be improved (data normality, analyze the interaction effect between topographic parameters and land use, MANOVA). In my opinion, some major aspects should be addressed before the acceptance of the manuscript for publication. The text

should be revised to improve the grammar and readability.

Line 13 "SOC levels" content should be more appropriate to refer the percentage of SOC in a soil sample. Please revise in the whole manuscript. See line 74. Keywords: "landscape" is too general Line 26 Human activities are responsible for . . . that exacerbates soil erosion. Please include a reference. Line 32 "Soil degradation is a key component of land degradation in the Mediterranean region" a similar sentence was included in line 25 "Land degradation, and especially soil degradation, is a major challenge for Mediterranean" Lines 37-41 could be move before line 32 Lines 42-45 can be moved to line 26 after ecosystems Line 45 OM should be defined Soil organic matter similarly SOC in line 52 Line 51 "Some soil biological degradation types were caused by agricultural practices" should be linked with "Soil degradation processes include biological degradation" in line 42 Line 52 Please include a reference also in lines 59 and 65 Line 54 remove "of OM" Line 60 a sign of Line 61 Soil from the study area Line 62 carbonate sedimentary parent material? Lines 63 – 67 This paragraph refers to Tunisia? It is unclear in the present form. Line 73 "systems" can be removed; Revise the use of landform that refers to slope and aspect (see line 340). Landform is more than derived topographic variables, it implies geomorphic processes. Similarly in line 77 environmental factors to refer slope and aspect. Line 75 The aim of this study is to quantify SOC content and evaluate the factors that affect the SOC variation, specifically, . . .. Line 81 In the introduction the problem statement can be related to Wadi Beja watershed; Authors should list the soil degradation problems in this area or why this watershed was selected to link with the research question (See lines 104 – 107; 111 - 114). Apart from the lack of soil database Line 100 m a.s.l Line 111 conserved? Line 139 What soil types? A description of the soil type or a map should be included. Line 142 Define LUC. Include the abbreviation in line 141. Please revise in line 152 LUS or LUC. Use the same abbreviation in the whole manuscript. Line 145 flat, moderate and steep A figure to illustrate the location of sampling units and sampling collection strategy (Lines 153-156) is needed. Also derived land use maps from Landsat in 1985, 2002 and 2016 could be included. Table 2 Slope (%) "categories" can be removed

Line 159-163 Move to introduction. Authors should put in value the novelty of male the first soil library Line 164 "All samples were taken for spectroscopy analysis in the laboratory." This sentence is unclear. Line 170 Determination of SOC. Please explain in detail SOC determination by dry combustion? Line 181 before applying ANOVA test, data normality should be verified. Lines 187-189 move to Materials and methods Results some of the sampling points have been affected by land use changes? Authors can consider using a multivariate ANOVA In the results section, authors included references (see lines 215, 217) to interpret the results. Please revise and move these parts to Discussion Lines 215 and 217 Delete the brackets Discussion Authors focused in land management strategies. However there is a lack of discussion regarding the impact of land use changes on SOC in the study watershed. Lines 283 – 286 This paragraph seem disconnected Conclusions should be shortened Lines 439-443 move to discussion

---

## Editor Comment (EC1) · Estela Nadal Romero (Editor) · 8 May 2019

Dear authors, although both reviewers indicated that your manuscript discusses a very interesting research topic, both suggested major revision due to incorrect explanation of the methodology, statistics analysis, confussion in results and discussion section. In addition both reviewers indicated that your manuscript should be reviewed by an english editor as there are many grammar and language mistakes.

---

## Short Comment (SC1) · 11 May 2019

Dear Sir

Thank you for your constructive revisions. Unfortunately, I couldn't download the full text of Thangavel Ramesh et al. (Advances in Agronomy, 2019). Please can you send me the paper? In the meantime, I am working in the revision, then I will send it again to Englich proof reading.

Many Thanks.

Donia Jendoubi.

---

## Author Comment (AC1) · 3 Jun 2019

Dear Reviewer 1

Thank you for your consttructive comments. I could address all the comments and and English proof reading was made after that. I am keeping 2 versions which a first one is with track changes showing all the corrections and a second one with proof reading changes.

Many Thanks.

[Figure]

Please also note the supplement to this comment:
https://www.soil-discuss.net/soil-2019-15/soil-2019-15-AC1-supplement.pdf

[Figure]

**Supplement:**

GENERAL COMMENTS The article discusses a very important topic and is based on extensive dataset. However, because of poor English and lack of logic in presentation of research, the reading is quite difficult. Moreover, there is a lack of adequate explanation of the research context (state-of-the-art revision and indication of the scientific gap to be approached and methodology. We would recommend the author to look at the recent review of the topic by Thangavel Ramesh et al. (Advances in Agronomy, 2019). What is the new knowledge this work contributes to that already published? The article considers "land forms" as synonym to slope and aspect, which is not true. There is no clear explanation of methodology; confusion in Results and Discussion sections, and, as a consequence, Conclusions are not supported by Results and Discussion. Incor-

rect citing format through the article (you cannot begin the sentence with parenthesis). Example: Line 210: "(Kravchenko et al., 2002 and Jiang and Thelen, 2004) found . . ." Line 212: "(Herrick and Wander, 1997) showed . . ."

Specific comments: English style is very poor. The article is full of English mistakes of all types. Some examples: Abstract (lines 13-14): "For permanent crops, which was interplanted . . .." – the verb was is incorrect because crops is a plural noun Line 35: ". . . soil quality was seen in relation to soil conservation in agricultural systems, which aims at sustaining the productive capacity of soils, . . ." – impossible Line 167: "The performance evaluation of the prediction model was created on the following statistical criteria: . . ." - impossible Line 211: "slope plays a great role on crop yield" – mistake in the use of preposition Line 217: "Forests showed significantly the highest SOC" – the weird word order

Methodology: No table with descriptive statistics of data. No details on data preprocessing, spectral measurements instrumentation and protocol. No explanation of classification procedure used for land use detection from Landsat images. The method applied for assessment of slope and aspect impact is not explained. It seems that only one-way ANOVA was used to assess the impact of slope and aspect on SOC. Application of multivariate statistical modeling would be more appropriate. One of the key experts in VIS-NIR spectroscopy is incorrectly cited two times (different errors): Line 171: "Vicsarra et al., 2006" Line 188: "Rossel et al. 2006)"

Figures:

Figure 1, the map should be improved, the gray backgound (topographic sheet) should be removed. Figure 5 and 6 contain redundant information.

---

## Author Comment (AC2) · 3 Jun 2019

Dear Reviewer 2

Thank you for your constructive comments. I could address all the comments and English proof reading was made after that. I am keeping 2 versions which a first one is with track changes showing all the corrections and a second one with proof reading changes.

Many Thanks.

[Figure]

Please also note the supplement to this comment:
https://www.soil-discuss.net/soil-2019-15/soil-2019-15-AC2-supplement.pdf

**Supplement:**

Soil-2019-15 is an interesting study which presents the combined effect of land use and topography on soil organic carbon (SOC) by comparing SOC data under four land uses and three slope categories and two aspects in north-western Tunisia. This paper contributes to build a soil spectral database of the studied watershed as a tool to study SOC variability. Although the consistent soil sampling collection and laboratory work, there is a lack of detailed information about materials and methods (soil type, soil spectral collection, model spectrometer, partial least-square regression) and the statistical analysis can be improved (data normality, analyze the interaction effect between topographic parameters and land use, MANOVA). In my opinion, some major aspects should be addressed before the acceptance of the manuscript for publication. The text

should be revised to improve the grammar and readability.

Line 13 "SOC levels" content should be more appropriate to refer the percentage of SOC in a soil sample. Please revise in the whole manuscript. See line 74. Keywords: "landscape" is too general Line 26 Human activities are responsible for . . . that exacerbates soil erosion. Please include a reference. Line 32 "Soil degradation is a key component of land degradation in the Mediterranean region" a similar sentence was included in line 25 "Land degradation, and especially soil degradation, is a major challenge for Mediterranean" Lines 37-41 could be move before line 32 Lines 42-45 can be moved to line 26 after ecosystems Line 45 OM should be defined Soil organic matter similarly SOC in line 52 Line 51 "Some soil biological degradation types were caused by agricultural practices" should be linked with "Soil degradation processes include biological degradation" in line 42 Line 52 Please include a reference also in lines 59 and 65 Line 54 remove "of OM" Line 60 a sign of Line 61 Soil from the study area Line 62 carbonate sedimentary parent material? Lines 63 – 67 This paragraph refers to Tunisia? It is unclear in the present form. Line 73 "systems" can be removed; Revise the use of landform that refers to slope and aspect (see line 340). Landform is more than derived topographic variables, it implies geomorphic processes. Similarly in line 77 environmental factors to refer slope and aspect. Line 75 The aim of this study is to quantify SOC content and evaluate the factors that affect the SOC variation, specifically, . . .. Line 81 In the introduction the problem statement can be related to Wadi Beja watershed; Authors should list the soil degradation problems in this area or why this watershed was selected to link with the research question (See lines 104 – 107; 111 - 114). Apart from the lack of soil database Line 100 m a.s.l Line 111 conserved? Line 139 What soil types? A description of the soil type or a map should be included. Line 142 Define LUC. Include the abbreviation in line 141. Please revise in line 152 LUS or LUC. Use the same abbreviation in the whole manuscript. Line 145 flat, moderate and steep A figure to illustrate the location of sampling units and sampling collection strategy (Lines 153-156) is needed. Also derived land use maps from Landsat in 1985, 2002 and 2016 could be included. Table 2 Slope (%) "categories" can be removed

Line 159-163 Move to introduction. Authors should put in value the novelty of male the first soil library Line 164 "All samples were taken for spectroscopy analysis in the laboratory." This sentence is unclear. Line 170 Determination of SOC. Please explain in detail SOC determination by dry combustion? Line 181 before applying ANOVA test, data normality should be verified. Lines 187-189 move to Materials and methods Results some of the sampling points have been affected by land use changes? Authors can consider using a multivariate ANOVA In the results section, authors included references (see lines 215, 217) to interpret the results. Please revise and move these parts to Discussion Lines 215 and 217 Delete the brackets Discussion Authors focused in land management strategies. However there is a lack of discussion regarding the impact of land use changes on SOC in the study watershed. Lines 283 – 286 This paragraph seem disconnected Conclusions should be shortened Lines 439-443 move to discussion

---

## Author Comment (AC4) · 3 Jun 2019

Dear R1

I am attaching the 2 versions here.

Many Thanks.

Please also note the supplement to this comment:
https://www.soil-discuss.net/soil-2019-15/soil-2019-15-AC4-supplement.zip

---

## Author Comment (AC5) · 3 Jun 2019

Dear R2

I am attaching the 2 versions here.

Many Thanks

Please also note the supplement to this comment:
https://www.soil-discuss.net/soil-2019-15/soil-2019-15-AC5-supplement.zip

---

## Author Comment (AC6) · 3 Jun 2019

Dear Editor

I am attaching the 2 versions here.

Many Thanks

Please also note the supplement to this comment:
https://www.soil-discuss.net/soil-2019-15/soil-2019-15-AC6-supplement.zip

---

## Author Response (AR2)

**Suggestions for revision or reasons for rejection (will be published if the paper is accepted for final publication)**

The manuscript SOIL-2019-15 "Impacts of land use and topography on soil organic carbon in a Mediterranean landscape (north-western Tunisia)" has been improved by the authors after having into account most of the comments made by the referees. In this new version, provided information concerning the study area and methodology and sampling design is much clear and understandable. However, still some aspects should be considered before being publishing. Above all discussion should be improved, in the present form is wordy and it is difficult to extract the meaning. Why Authors did not include a brief discussion about the 3.1 soil spectral library as an integrative indicator of soil quality. This is one of the novel aspects of the article.

Please avoid the repetition of the information. Be careful to keep verb tense consistent within sections of a paper. For example, the Results section is usually in the past tense (because the experiments have already been done).

• Please keep the same terminology in the whole manuscript; choose among topographic characteristics or features (instead of topographic units or factors). Similarly SOC levels or contents.

• The acronyms should be defined for the first time. Revise in line 8 soil organic carbon (SOC) and line 51 Soil organic matter (OM); Line 84 SOC instead of soil organic carbon

• Please some concepts should be rewritten to be more accurate Lines 29-30 change people by human activities … exacerbates soil erosion". Lines 31- 32 Long-term anthropogenic pressure from agricultural use (Kosmas et al., 2015) together with abiotic factors such as climatic and topographical variability.

• A deeper English proofreading can improve the readability of the text. Please remove to avoid the repetition of information Line 35 "Soil degradation is a key component of land degradation in the Mediterranean region" a similar sentence was included in line 28. In addition, revise to avoid the repetition of the same word e.g. "soil quality deterioration contributes to the deterioration". See also lines 89-90

• It is not clear if Authors refers to north-western Tunisia (lines 67-73, line 101)

• Lines 77, 360 Include a reference

• Line 83 soil organic carbon levels?

• Lines 128-129 ranges of slopes corresponding to high, steep and moderate can be included. These categories are the same as in table 2?

• Line 185 Soil type should be included of the study soils

• Figure 2 The 24 different units can be delineating

• Table 2 include aspect for steep category. "in" can be removed (in %). The exact number of soil samples per unit should be included in Table 2 (see lines 155 – 156) by including land use

• Lines 219 and 251 31%

• Line 203 This sentence from the previous version of the manuscript "Soil samples were air dried (to 30 °C) and sieved to pass through a 2 mm mesh." Should be included.

• Please avoid the repetition of information by deleting lines 206-207 (info repeated in lines 199-200); lines 207-213 (repeated in lines 200-201 and figure 2); line 245 (info is provided in lines 232-233); line 268 (info is included in line 271); Lines 418 – 423 (info is repeated in lines 394-409)

• Line 253 (vario MICRO tube, Elementar)

• Line 258 This study is focused on SOC then this sentence can be omitted.

Did Authors check that TC measurements and SOC measurements are coherent? TC >> SOC

Authors measured SIC by other method? To control / verify their SOC estimates?

• Line 342 – 350 Please revise the readability for a fluent English style. Similarly in the abstract. See the repetition of decline in two continuous sentences (lines 344-345)

• Lines 366 -372 This is not a discussion of the results. This paragraph presents materials and methods and objectives. Please remove because this information was included previously.

• Line 387 "enhanced the SOC content" instead of improved

• Line 388 appropriate land management instead of good land management

• Line 389 "as it reduced both the area covered only by very old cereal monocultures and the soil degradation." This sentence is unclear in the present form.

• Line 405 assuming that forest soils are not affected by soil erosion because their higher SOC contents. I do not agree with this statement.

• Line 409 Not only low erosion rates please include other key factors for high SOC content in forest soils: higher carbon input, low litter decomposition …

• Lines 410-417 This general information about Mediterranean soils should be move to the beginning of the discussion

• Lines 420 – 421 Please revise and rewrite, the sentence in the present form is unclear

• "In lands where field crops once were, if they have been interplanted with permanent crops, the SOC content has improved"

• Line 421 "Overgrazing and bad management of grazing lands has led to SOC decreases." Please move to line 412 and link with inappropriate land management

• Line 422 Remove "Finally, forest land use has the highest SOC content, as it is protected by forest regulation and less disturbed." This info is included in line 409 "which is highly related to the lower disturbance in the forests"

• Line 428 "Different soil properties encountered among" Which soil properties have been determined by authors apart from SOC. This information should be very useful to include a description in materials and methods about soil characteristics and soil type

• To say "undoubtedly have effects on SOC content" these key soil properties should be included

• Line 436 in soils

• Line 462 I doubt about this statement: "This can be explained by the fact that soils on flat slopes tend to be thicker as a result of deposition". Have authors any evidence from the results?

• Line 476 nutrients loss instead of nutrition loss

• Lines 478-483 This information is repeated and discussed previously in Discussion section

• Lines 392 and Lines 547 – 552 Since the work is not focused on C sequestration and it is not introduced previously. The main topic was SOC variability as soil quality parameter and its controlling factors (topography and land use). I recommend remove this paragraph.

[revised manuscript text omitted]

---

## Author Response (AR3)

**Suggestions for revision or reasons for rejection (will be published if the paper is accepted for final publication)**

Manuscript has been improved considerably and a great effort was made by Authors. Please see below some comments:

• A deeper English proofreading can improve the readability of the text.

• Lines 42-44 Agricultural practices is not the only factor for soil degradation processes (biological, physical, chemical). A more generic statement could be included: Soil degradation processes comprises or include biological degradation … chemical degradation (References).

Also revise brackets "(e.g. acidification nutrient and depletion (Diodato and Ceccarelli, 2004; Post and Kwon, 2000)" I suggest move the reference at the end of the sentence.

• Lines 112-113 Information repeated in lines 113—116 please delete (this suggestion was included in the previous report) "In this study, we explore SOC distribution according to land use across topography (slopes and aspects) in north-western Tunisia".

See lines 113-116: The aim of this study is to quantify SOC content and evaluate the factors that affect SOC variation, specifically the mechanisms affecting differences in SOC distribution patterns along different land use systems and topographic features (slope and aspect) in a Mediterranean ecosystem dominated by agricultural activities in north-wester Tunisia.

• Lines 169-170 Please include the ranges of slopes corresponding to steep (>16%), moderate (8% – 16%) and flat (0% – 8%) should be included. (this suggestion was included in the previous report)

Line 170 Correct high to steep? Only three categories are included in Table 2: steep, moderate and flat. High?

• Table 2 include aspect for steep category.

• Line 415 north-facing areas whereas 1.14% of SOC ….

• Line 564 lower disturbance in forest soils by erosion?

• Line 569 The accumulation of SOC variation? Authors mean: High SOC variation on hillslopes?

• Line 600 "the highest"

[revised manuscript text omitted]